# Group Emotion Detection Based on Social Robot Perception

**DOI:** 10.3390/s22103749

**Published:** 2022-05-14

**Authors:** Marco Quiroz, Raquel Patiño, José Diaz-Amado, Yudith Cardinale

**Affiliations:** 1Electrical and Electronics Engineering Department, School of Electronics and Telecommunications Engineering, Universidad Católica San Pablo, Arequipa 04001, Peru; marco.quiroz@ucsp.edu.pe (M.Q.); rpatino@ucsp.edu.pe (R.P.); jose_diaz@ifba.edu.br (J.D.-A.); 2Instituto Federal da Bahia, Vitoria da Conquista 45078-300, Brazil; 3Higher School of Engineering, Science and Technology, Universidad Internacional de Valencia, 46002 Valencia, Spain

**Keywords:** social robots, emotion detection, group emotion, group detection, facial expression recognition, group behaviour recognition, human–robot interaction

## Abstract

Social robotics is an emerging area that is becoming present in social spaces, by introducing autonomous social robots. Social robots offer services, perform tasks, and interact with people in such social environments, demanding more efficient and complex Human–Robot Interaction (HRI) designs. A strategy to improve HRI is to provide robots with the capacity of detecting the emotions of the people around them to plan a trajectory, modify their behaviour, and generate an appropriate interaction with people based on the analysed information. However, in social environments in which it is common to find a group of persons, new approaches are needed in order to make robots able to recognise groups of people and the emotion of the groups, which can be also associated with a scene in which the group is participating. Some existing studies are focused on detecting group cohesion and the recognition of group emotions; nevertheless, these works do not focus on performing the recognition tasks from a robocentric perspective, considering the sensory capacity of robots. In this context, a system to recognise scenes in terms of groups of people, to then detect global (prevailing) emotions in a scene, is presented. The approach proposed to visualise and recognise emotions in typical HRI is based on the face size of people recognised by the robot during its navigation (face sizes decrease when the robot moves away from a group of people). On each frame of the video stream of the visual sensor, individual emotions are recognised based on the Visual Geometry Group (VGG) neural network pre-trained to recognise faces (VGGFace); then, to detect the emotion of the frame, individual emotions are aggregated with a fusion method, and consequently, to detect global (prevalent) emotion in the scene (group of people), the emotions of its constituent frames are also aggregated. Additionally, this work proposes a strategy to create datasets with images/videos in order to validate the estimation of emotions in scenes and personal emotions. Both datasets are generated in a simulated environment based on the Robot Operating System (ROS) from videos captured by robots through their sensory capabilities. Tests are performed in two simulated environments in ROS/Gazebo: a museum and a cafeteria. Results show that the accuracy in the detection of individual emotions is 99.79% and the detection of group emotion (scene emotion) in each frame is 90.84% and 89.78% in the cafeteria and the museum scenarios, respectively.

## 1. Introduction

Social robots are increasingly being incorporated into crowded human spaces, such as museums, hospitals, and restaurants, in order to offer services, perform tasks, and interact with people. Social robots are considered as physical agents with the abilities to act in complex social environments [1]. They must imitate the socio-cognitive abilities of humans and explore behaviours to be empathic and aid with the interactions between robots and humans [2,3], which in turn demands more efficient and complex Human–Robot Interaction (HRI) designs. HRI must include behavioural adaptation techniques, cognitive architectures, persuasive communication strategies, and empathy [4].

A strategy to improve HRI is to provide robots with the capacity of detecting the emotions of the people around them in order to plan and organise different future actions, such as adapting behaviour, planning, navigation, and control. In this sense, visual perception could give them information to understand and recognise emotions, for example through the user’s body language and vocal intonation [5] or through facial expression [6,7]. According to the detected emotion and the specific situation, robots adapt their actions to show appropriate behaviours.

In a single HRI system, it is still complex to consider all the methods and approaches of emotion recognition. Emotional systems are limited to recognising emotions in specific situations and in controlled settings. The main challenges that these systems must overcome are the flexibility of the algorithms to adapt to real environments (dynamic environments), the consideration of the inter-cultural variations of people, the detection of groups of people, and the recognition of group emotions [8,9,10].

In a group, people can express different emotions, and the robot must process the emotion of each person and summarise them in a group emotion to define its actions. In such a case, it is necessary to consider the robots’ first-person perspective of the world. Cameras mounted on the robots’ head or chassis have allowed studying the scenes from a point of view that provides robots such a first-person perspective of the world. This field of research in computer vision is known as egocentric or first-person vision [11]. This practice is useful when the social robot interacts with more than one person, for example in social environments, such as schools, hospitals, restaurants, and museums.

The third-person camera is a device outside of the robot. Egocentric vision presents advantages in comparison with the third-person camera as the robot is recording exactly what it sees in front of it; the camera movement is driven by the robot’s body, and the stabilisation of the image is controlled by the robot itself. Robots can use egocentric vision to recognise emotions, navigate, or detect different objects. Developing systems with this perspective makes the robot able to adapt to social groups of humans [12]. The detection of groups of people improves the navigation of a social robot in indoor and outdoor environments, and the detection of group emotions allows the robot to improve HRI, exhibiting acceptable social behaviour [13,14,15,16], as well as associating the group emotion with the scene in which the group is participating. Nevertheless, most existing studies related to detecting group emotions are based on third-person cameras [17,18,19,20,21], but their complexity makes them unsuitable for social robots with egocentric vision due to their sensory capacity.

In this context, to overcome some limitations of existing studies, a system to recognise scenes in terms of groups of people, to then detect the global (prevalent) emotion in the scene, is proposed in this work. The scene detection is based on the size of the faces of people detected by the robot during its recognition process; from the robot perspective, the size of faces decreases when the robot moves away from the group of people. In each frame, individual emotions are first recognised through a Visual Geometry Group neural network to recognise faces (VGGFace (https://www.robots.ox.ac.uk/, accessed on 1 May 2022)), to then identify the frame emotion, and finally, detect the emotion of each scene. Additionally, in the absence of adequate datasets with a robocentric perspective for the training and validation processes of the machine learning models used, a strategy is proposed for the creation of a dataset with images, to validate the estimation of individual emotions, and a dataset with videos, to evaluate the detection of scenes and the emotions of these scenes. Both datasets were generated from videos captured by robots through their sensory capabilities, in a simulated environment with the Robot Operating System (ROS) (https://www.ros.org/, accessed on 1 May 2022).

To evaluate the efficiency of the proposed approach to recognise the emotions of groups of people conforming scenes, several experiments were carried out with two simulated scenarios in ROS/Gazebo: a cafeteria and a museum. With the implementation of the proposed system, the robot is endowed with new capabilities for the perception of the emotion of an environment, based on the emotion of various scenes, which in turn allows improving the interaction between the robot and people.

To show the entire process of this work, the rest of this document is organised as follows. In Section 2, studies related to the detection of group emotions are explained. Section 3 explains the proposal for the detection of individual emotions, emotions per frame, and emotions per scene. To validate the detection of these emotions, in Section 4, a method to create a database is presented. Section 5 shows the implementation of the proposal in Gazebo and ROS and shows the results obtained in these simulations. Discussions about the findings are presented in Section 6. Finally, Section 7 presents the final conclusions and future research.

## 2. Related Work

Most proposals related to this research are focused on the recognition of group emotions and the study of the effects this has on the planning and behaviour of social robots. Regarding the first aspect, some recent and relevant studies, although they are not in the robotics area, are reviewed. For social robots, only a few works dealing with group emotion recognition were found, which are described afterwards.

### 2.1. Group Emotion Recognition

To analyse the proposals that perform the recognition of group emotions, four aspects are considered: pre-processing of the images, feature extraction, the fusion method, and evaluation. The review focus is on works that mainly base the recognition of emotions on facial expressions, as is done in this work.

#### 2.1.1. Pre-Processing of Images

To estimate the emotions of a single person or of groups of people on images, a pre-processing step is demanded to detect the regions of interest, which can be faces, bodies, or other objects within the image that influence the emotion.

For face detection, some approaches are based on neural networks, such as the Multitasking Convolutional Neural Network (MTCNN) [22]. This network uses three cascaded convolutional networks to improve face detection accuracy, which makes its use very common [18,23,24,25,26,27,28,29]. There are other methods that also use neural networks, such as RetinaFace [21,30], PyramidBox [19], TinyFace [19,20,31], and the Single-Shot Scale-Invariant Face Detector (S3FD) [32]. Other methods do not use neural networks for face detection, such as the Viola–Jones algorithm [33], which uses Haar characteristics to locate the face in an image. This technique was used in the study presented in [17] and in the work described in [34]; it was used in conjunction with the Histogram of Oriented Gradients (HOG). In [35], Seetaface was used, an algorithm in which there are several cascading classifiers. A mixture of trees to detect faces and postures was used in [36,37]; this model detects faces even when a deformation exists due to a facial expression. Once the face is detected, additional steps can be performed, such as face frontalisation [38], to make all detected faces have a frontal orientation through matrix projection or determine the importance of each face in a group through Cascade Attention Networks (CANs), as proposed in [25].

To estimate a global emotion in a group, other aspects different from faces and bodies are considered, fr example detecting areas containing salient objects or features that can influence the emotion of the group [18]. In [26], the removal of faces was performed, using heat maps with Gaussian distributions, to obtain a cleaner representation of the scene.

#### 2.1.2. Feature Extraction

As mentioned in the previous section, the detection of group emotions is carried out through the face, posture, skeleton, visual attention (i.e., points of interest of members of the group), and the elimination of faces to consider only objects in the environment (i.e., the context). To detect group emotions according to these characteristics, it is common to use different models of neural networks to process each modality (face, posture, context, etc.), in which different feature extraction architectures are used.

Most common architectures for face feature extraction are based on neural networks, the VGGFace neural network [39] being the most used for the extraction of face features, as done in [17,23,25,28,29,31,34,35]. This neural network was trained with 2.6 million images, and its main function is face recognition. To improve the feature extraction process, VGGFace can be used in conjunction with other architectures, such as Squeeze-and-Excitation Networks (SENets), Residual Networks (ResNets), Deep Convolutional Neural Networks (DCNNs), and Graph Neural Networks (GNNs), to improve accuracy in estimating individual emotions, as followed in [18,25,28,35]. There is another version of this neural network, known as VGG2-Senet-ft-FACE (pre-trained with the VGGFace2 database), which results from the combination of the ResNet and SENet networks, as described in [18], but it can also be used separately, as the study in [20,25] proposed.

Residual networks are used to extract characteristics from the facial region using the ResNet-18 neural network in [32]. In [24], two residual networks of different ResNet-64 layers (to process aligned faces) and ResNet-34 (to process non-aligned faces) were used. In [26], two ResNet models were used: ResNet-18 for small faces (size less than 48 × 48) and ResNet-34 for large faces (size larger than 48 × 48). To improve the precision in the detection of individual emotions, apart from using a Dense Convolutional Network (DenseNet201), two neural networks (Inception-ResNet-v2) can be combined, as in [19], or new blocks (e.g., excitement and comprehension blocks) can be added to a neural network, as in [18,25].

Since the input is low quality images, in [36], a reduced AlexNet architecture was used for feature extraction, in which the input image was cropped to 40 × 40 pixels. To predict facial emotions, the study presented in [37] tried a pre-trained Convolutional Neural Network (CNN) and a CNN trained from scratch; the best results were obtained with the pre-trained model. Similarly, in [27], several CNNs with different depths were used, but in this case using the softmax angular loss (A-Softmax) to make the learned characteristics more discriminative. In [40], after detecting the faces, the neural networks VGG-16 and MobileNet-v1 were used to extract the characteristics of each face. Instead of training a neural network, in [21], EmoNet was proposed. This architecture improves the convolutional operator, increases the capacity of the network, and reduces the spatial dimension in the first layers.

To recognise group emotions in a video, static and spatial-temporal characteristics were considered in [30]. Static features were used to estimate the emotion on each frame of the video from individual faces and postures, while spatial–temporal features considered both audio and video. To extract face and posture characteristics, CNN, Batch Normalisation Inception (BNInception), and ResNet models were respectively used.

#### 2.1.3. Fusion Methods and Evaluation

Once individual emotions are detected, a fusion method is applied to estimate the group emotion. The most common fusion method are the weighted sum, in which a weight is assigned to each score, according to the size of the face, for example, as used in [17,18], and the average scores, used in [23,24,26].

More sophisticated fusion methods can also be used, such as neural networks. The Long Short-Term Memory (LSTM) neural network was used in [35,36,40] to learn how individual emotions affect the group emotion. Residual networks, such as cascade attention networks, were used in [25,32], to determine the influence of each face in the detection of the emotion of the group.

Other less-popular fusion methods have been used, such as attention mechanisms, used in [27], the Frame Attention Network (FAN) model, proposed in [30], and the combination of feature vectors, as in [19]. Similarly, in [29], the three feature vectors (scene, face, and object) were concatenated and weighted to learn the weights of the context-aware fusion. Attention mechanisms use the individual face feature vectors to predict the group emotion. In [27], several attention mechanisms were tested, in which the best results were obtained with a fully connected neural network combined with a weighted sum. In [19], the average, minimum, and maximum feature vector were concatenated to train a Multilayer Perceptron (MLP) to determine the group emotion. In [30], to determine the emotion in videos, the authors used a Frame Attention Network (FAN), composed by a function incorporation module and a frame attention module, to generate a single feature vector. In [20], the Discrete label to Continuous score (D2C) method was implemented to estimate group cohesion scores considering the interaction between continuous and discrete labels. In [21], a different fusion method was proposed, called Non-Volume Preserving Fusion (NVPF). This method stacks the features of each face to form a single group-level feature and then models a probability density distribution to account for the individual and group-level features.

Concerning the evaluation metrics to show and compare the results, the studies use the following metrics: accuracy, Mean Absolute Error (MAE), Root-Mean-Squared Error (RMSE), and Mean-Squared Error (MSE), the accuracy being the most used one.

#### 2.1.4. Comparative Evaluation

All these works are summarised in Table 1, emphasising group emotion detection from faces, considering pre-processing, individual emotion detection, and the fusion method. These works demonstrate that neural networks have been successfully used for face detection, with MTCNN models, and for feature extraction to detect individual emotions, with the VGGFace architecture. It is also worth noting that ResNet architectures are also common for feature extraction. The fusion methods used seem to be more variable, with a light trend of using attention mechanisms and the average of individual emotions.

### 2.2. Emotion Recognition for Social Robots

In the context of HRI, emotion recognition has become an essential strategy to generate the behaviours of social and service robots sharing spaces with humans. According to the emotion detected, the robot can modify its behaviour or its navigation, showing a socially accepted attitude. The study presented in [41] described 232 papers focused on emotional intelligence (i.e., how the system processes the emotion, the algorithm used, the use of external information, and the alteration of emotions based on past information), the emotional model, or the implementation of the model, showing the trends and advancements of improving HRI from these three perspectives. The authors in [9] mentioned the importance of emotion recognition for HRI.

Robots expressing emotions are also another aspect of interest in this area, as shown in [42]. That survey presented a review of research papers from 2000 to 2020 focused on studying the generation of artificial robotic emotions (stimulus), human recognition of robotic artificial emotions (organism), and human responses to robotic emotions (response), as a contribution to the robotic psychology area. These works described in both surveys [41,42] demonstrated that social robotics is a growing area, where psychology and sociology aspects converge [8].

The estimation of individual emotion also influences the proxemic behaviour that a social robot should have. This separation between the robot and people can be limited by the accessibility distance, the user’s comfort distance, and the user’s emotion. Based on these features and the ability of robots to recognise moods or emotional states of people, robots can plan the best routes to follow [15,43,44].

A variety of sensorial capacities allows robots to capture several multimedia contents (e.g., images, videos, speech, text), from which emotions can be detected. As in this work, many studies are focused on face emotion recognition from images and videos to improve HRI or social navigation. A survey of 101 papers from 2000 to 2020 dealing with the detection of human facial emotion and generation of robot facial expressions was presented in [45]. The authors compared the accuracy of face emotion recognition from images in the wild versus images in controlled scenarios, revealing that for the first case, the accuracy was considerably lower than for the second case. As an effort to improve the accuracy when the information is taken from the wild (as for social robots in service), an emerging strategy consists of considering multimodal or multisource approaches. Thus, a few works have started to adopt multimodal approaches combining several modalities based on the information captured by several robots’ sensors, such as: (i) from Kinect cameras to recognise emotion based on human facial expression and gait, as the study presented in [46]; (ii) from cameras and the speech system of robots, some studies combine facial and speech [47,48,49,50,51,52,53] and body gesture and voice [5] to detect human emotions and accordingly improve HRI or navigation; (iii) from text and speech by converting speech to text to then apply Natural Language Processing (NLP) to recognise emotions, as done in [54]. However, this topic of robotics is still limited, as the survey presented in [55] reported.

Concerning group emotion recognition in social robotics, only a few studies dealing with group detection and recognition of individual emotions were found. For the navigation of social robots, parameters such as the trajectory, position, or speed of the movements of people or the robot itself were considered, but they did not take into account the emotions of multiple people [12,56,57,58]. There are studies that consider the influence of a robot within a group of people [13,16,59], but the detection of group emotions was not carried out and even less the detection of the emotion of an environment. There are very few studies proposing methods for group emotion estimation. In [60], based on individual emotion recognition with a Bayesian network, an approach to estimate group emotion from face expressions and prosodic information was proposed. Similarly, with a Bayesian network and individual facial expression recognition, but combined with environmental conditions (e.g., light, temperature), in [61], an approach to estimate the group emotion to then produce appropriate stimuli to induce a target group emotion was presented. Furthermore, from individual facial expressions, in [62], a system to recognise group emotion for an entertainment robot was described. In [63], research on HRI in small groups was carried out, concluding that groups are complex, adaptable, and dynamic systems. The authors recommended developing suitable robots for group interactions and improving the methodologies used in the process of measuring human and robotic behaviour in situations involving HRI.

Without pretending to be an exhaustive review, these studies revealed that some limitations and some challenges are still open in the area of HRI and social navigation considering groups of people. Even though emotion recognition for social robots has become the focus of many works and presents important advancements, group recognition, group emotion recognition, and even scene emotion detection are still the first steps, leading to the lack of available datasets to support the training, testing, and validation of machine learning models to do so. The RICA database [64], generated from a robocentric view, has been presented, but it is focused on group interactions. In this work, a new approach to estimate group emotions from a robocentric perspective is proposed. With the proposed approach, robots are able to detect groups of people conforming a scene and estimate the scene emotion. To do so, the proposed approach is based on classical machine learning models as those shown in Table 1. For pre-processing, the Viola–Jones algorithm is used; to extract face features and detect individual emotions, the VGGFace neural network is used and the average as the fusion method; all of these were adapted to be performed in the embedded hardware of robots. Since group emotion detection is just currently emerging, there is a lack of appropriate datasets for training and validation; thus, a strategy to create datasets with videos and images taken from the sensor capacity of robots, in simulated environments, is also proposed.

## 3. Group Emotion Detection: The Proposal

The proposed emotion detection approach is based on a machine learning model focused on the analysis of individual emotions and the size of faces, to identify a scene conformed by a group of people and then recognise the emotion of such a group (scene emotion). To better understand the whole pipeline of the proposal, the definitions of different elements from the point of view of the robot are first presented, as follows:Video (*V*): This is a recording of a sequence of images (frames) of an indoor space, taken from the robot sensors. While the robot is moving around the room, it records what it *sees*, with the aim of detecting groups of people (scenes).Frame (*f*): This is an image of the set of images in a video. In this case, the frames with people are the targeted frames for the robots, in which an emotion is recognised, denoted as f.emotion.Scene (*s*): This is a sequence of frames (short video) in which a group of persons is detected. In each scene, an emotion is recognised, denoted as s.emotion.Blocks of frames in a video (BOF): A video is divided into blocks of frames (BOF), each one conformed by β frames. The β parameter is provided by the users and defines the windows to identify scenes (i.e., the number of frames that a robot should analyse to detect scenes in the video). If the video has *n* frames, the video is divided into *k*BOF, where $k=\frac{n}{\beta }$; hence, $BOF_{j}=\left\{f_{1,j},f_{2,j},\ldots ,f_{\beta ,j}\right\}$, where 1≤j≤k and $f_{i,j}$ is the frame *i* of $BOF_{j}$.Set of biggest faces per BOF (BF): For each $BOF_{j}$, the area of the biggest face among the frames in $BOF_{j}$ is extracted, such that $BF=\{bf_{1},bf_{2},\ldots ,bf_{k}\}$, where $bf_{j}$ is the area of the biggest face found in $BOF_{j}$.A $BOF_{j}$ can contain two scenes at a maximum, since the start of a scene is marked by the $bf_{j}$ (the biggest face in $BOF_{j}$) if the $bf_{j+1}$ is smaller than $bf_{j}$, or several BOF might belong to the same scene, if $bf_{j}$ is smaller than $bf_{j+1}$. Hence, a video contains one or more scenes, such that $V=\{s_{1},s_{2},\ldots ,s_{u}\}$, where $s_{i}$ is the scene *i* in the video *V* and 1≤u≤2k.

The complete pipeline of the proposed approach is shown in Figure 1. First, the capture of frames is carried out through the front camera of the robot, while it navigates in the indoor space. On each frame with people, all the faces are detected by the Viola–Jones algorithm, and they are stored in a vector. For each stored face, the area of the face is calculated, the feature extraction is performed, and the individual emotion of each face in the frame is estimated. Then, the frame emotion is determined with a fusion method of individual emotions; if there is only one person in the frame, the emotion of the frame is the emotion of that person.

To detect a scene, the user has to set the value of β. If β is to low, the robot could detect many scenes that do not correspond to the real groups. In contrast, if it is too big, the robot might assemble several groups in just one scene. Therefore, the value of β has to be adjusted according to the scenario in which the robot participates, the density of people in the room, etc. Thus, the robot analyses blocks of β frames (BOF), by identifying the biggest face of each BOF and building the set of biggest faces ($BF=\{bf_{1},bf_{2},\ldots ,bf_{k}\}$). Afterwards, the robot compares if $bf_{j}\leq bf_{j+1}$ (where $bf_{j}$ is the largest face found in $BOF_{j}$), then all frames between $bf_{j}$ and $bf_{j+1}$ belong to the current scene, and the comparison continues; otherwise, those frames belong to another scene, and the frame containing $bf_{j}$ is the limit of the scene (this means that the frame that contains $bf_{j+1}$ has a face that is far away from the current scene (group) and should belong to another scene.

Finally, to determine the emotion of the current scene $s_{j}$, the emotions of the frames that belong to $s_{j}$ are used. The predominant emotion in these frames is the fusion method that determines the emotion of $s_{j}$. The output of this process is an image for the scene detected, highlighting the identified faces, the emotion of each face, and the emotion of the scene textually indicated, as shown in Figure 1. The process continues for all frames in the video (until the robot stops capturing images), and the final output is the emotion of the video. Each phase of the proposed pipeline is detailed in the following subsections.

### 3.1. Face Detection

The objective of this stage is to detect all the faces in a frame. For this, the Viola–Jones classifier is used as the face detector, which stores the coordinates of the upper left corner (x,y), the width (*w*), and the height (*h*) of each face. With the values of *w* and *h*, the area (w*h) of each face is calculated, whose information is used for scene detection. In such a case, as the robot moves towards a group of people, the area of the faces captured by the robot begins to grow, and when it moves away, the areas begin to decrease. With this information, the limits to determine a scene are established, which is explained in Section 3.5.

### 3.2. Feature Extraction

At this stage, the characteristics of the detected faces are extracted. The objective of this stage is to find a vector of characteristics that efficiently represents the useful information of the detected faces. This process is important because it reduces the amount of data that represent a face without loss of information. For this, the VGGFace neural network is used, pre-trained with 2.6 million images. The vector of image characteristics is in the flatten layer, where the multidimensional data (obtained by the convolutional layers) is transformed to one-dimensional data. According to the configuration of this neural network, the input images must have a size of 224 × 224 pixels.

### 3.3. Estimation of Individual Emotion

VGGFace was trained to recognise 2622 classes. However, in this case, there are not 2622 emotions to classify; therefore, the fully connected layers of the VGGFace model are modified, shown in Table 2. Layers fc6 and fc7 have 512 nodes and layer fc8 has 6 nodes, which represent the emotions to be classified (happy, sad, angry, fear, disgust, and surprise). In addition, a dropout of 0.5 was added, to reduce the overfitting of the neural network (layers d1 and d2). Once this setup is done, only the fully connected layer of the VGGFace neural network is trained with the image dataset. The weights of the convolutional layers (Section 3.2) are sufficient to extract the features of the simulated faces used in this work.

### 3.4. Estimation of Emotion in Each Frame, Scene, and Video

The representation of individual emotions in this work is categorical; thus, the six basic emotions proposed by Ekman [65] were considered in this work. Even though there are other approaches that consider other emotions as basic [66,67], most studies discussed in Section 2 used the Ekman representation.

A frame of a video is an image, in which different faces can be detected and, therefore, different emotions can be detected. The same is true for a scene, which is conformed by several frames, and a video, which in turn is a set of scenes. The fusion method used in this work to obtain the emotion of a frame, a scene, and a video is to consider the predominant emotion in each case and classify it according to three categories: positive emotion, neutral emotion, or negative emotion.

The reason for adding three additional categories to the six basic emotions is related to using the valence dimension (it indicates how negative or positive an emotion is). Table 3 shows the classifications of the six emotions into three categories. Surprise is considered a neutral emotion because it can be positive or negative. With respect to the other emotions, by default, these can be intuitively classified as positive or negative. In addition, positive and negative emotions will have a greater weight over neutral emotions.

### 3.5. Scene Detection

The duration of a scene, in which a group is detected, is determined from the area of identified faces in the frames. As the robot approaches or moves away from a group of people, the area of the faces increases or decreases, accordingly. In this approach, the robot analyses β frames to distinguish a scene. An example is shown in Figure 2, with β=10; every 10 frames conform a BOF, in which the robot extracts the biggest face among the frames belonging to that BOF (blue dots in Figure 2); since the area of the biggest faces in $BOF_{1}$, $BOF_{2}$, $BOF_{3}$, and $BOF_{4}$ keeps growing, all these frames belong to the first scene (until Frame 32 in $BOF_{4}$; green line in Figure 2); this is the limit of the first scene; from that frame until the end of $BOF_{6}$ conform the second scene. Thus, a scene is made up of all the frames that the robot captures while approaching a group (increasing face areas), and if the robot sees a far away face (decreased face area), it is considered as another scene.

Algorithm 1 details the detection of scenes and emotions in a video. The inputs of the algorithm are *V*, the video; β, the number of frames to process to determine a scene; and *n*, the number of frames in the video *V*. This algorithm returns *S*, which is the list of scenes in the video, each one with the emotion recognised; *F*, which indicates the start and end frames for each scene; and V.emotion, the emotion of video *V*. From β and *n*, the algorithm determines *k*, the number of Blocks of Frames (BOF) (Line 2). The algorithm goes through all the frames in each block $BOF_{j}$ to determine its emotion ($f_{l,i}.emotion$),and in each block $BOF_{j}$, the largest face is detected ($bf_{j}$) (Lines 3 to 8). Once all biggest faces in all BOF are determined, the algorithm proceeds to identify the scenes (Lines 9 to 17). If $bf_{j}<b_{j+1}$, the scene does not change ($s_{s}$), but if $b_{j}>b_{j+1}$, the scene changes ($s_{s+1}$): the scene $s_{s}$ would be made up of the frames $f_{init}$ and $f_{end}$. Thus, the frame where the biggest face ($bf_{j}$) is located is the first frame of the scene $s_{s+1}$. Finally, the emotion of each scene is determined (Lines 18 to 20), as well as the emotion of the video (Line 21).
**Algorithm 1:**Scene and Emotion Detection for a Video**Input:***V* = the video; β = number of frames per block; n = number of frames in the video.**Output:***S* = array of scenes, *F* = pairs of frames that delimit each scene.1.  s=1, $f_{init}=1$ // First scene, first frame of the first scene.2.  $k=\frac{n}{\beta }$ // Number of BOF to analyse.3.  **for** i←1**to***k* **do**4.   **for** l←1**to**β **do**5.    $f_{l,i}.emotion\leftarrow$ determine the emotion of the frame6.   **end for**7.   bf[i]← determine the largest face of block $BOF_{i}$.8.  **end for**9.  **for** j←1**to**k−1 **do**10.   **if** bf[j]>bf[j+1] **then**11.    S[s]← add a new scene $s_{s}$.12.    $f_{end}\leftarrow$ frame where bf[j] is located.13.    $F\leftarrow [f_{init},f_{end}]$ // first and last frame that form the scene S[s].14.    $f_{init}\leftarrow$$f_{end}$15.    s←s+116.   **end if**17.  **end for**18.  **for** j←1**to**size(S) **do**19.    $S.s_{j}.emotion\leftarrow$ detect scene emotion $s_{j}$ in *S*20. **end for**21. V.emotion← determine emotion of video *V* from *S*22. **return V.emotion, S, F** // Video and scenes detected with their emotions tagged.

## 4. Generation of Datasets

As shown in Section 2, there is increasing research in the development of methods for the detection of group emotions and competitions, such as EmotiW (https://sites.google.com/view/emotiw2020, accessed on 1 May 2022), which help in the development of this research. In spite of these advances, the work related to the detection of group emotions from a robotic perspective is not very common. Consequently, as far as we know, there are no databases containing images or videos taken by robots (i.e., from an egocentric view). Therefore, with the help of ROS and Gazebo, we created datasets (https://github.com/marco-quiroz/Dataset-in-ROS, accessed on 1 May 2022) to validate the results obtained with the proposed method. The social robot Pepper simulated in ROS/Gazebo was used, which has various sensors to know its environment; in this case, only the front camera located in the upper part of the robot’s head was used. This camera has a resolution of 640 × 480 pixels at a speed of 1–30 fps.

The methodology to generate the datasets is shown in Figure 3. With this methodology, two datasets were generated, the first one made up of images and the second one made up of videos. The images dataset was used to train and evaluate the modified VGGFace neural network, while the videos dataset was used to detect scenes, as well as to validate the emotions of each frame and each scene. In the first stage of the methodology, the virtual environment was generated (e.g., museum, cafeteria, office), where the formation of groups was performed. The virtual environments where the simulations were carried out were created by the ROS community. For the images dataset, all the virtual characters (i.e., person representations in the environment) have the same emotion, since the idea is to have different faces, but with the same emotional expression. This is followed by a video recording of the robot’s path. Face detection is performed on each video. The labelling of the images dataset is automatic, because all the detected faces have the same emotion. For the videos dataset, the groups are conformed by persons with different emotional expressions, as in a scene, people can express different emotions. In this case, the labelling is performed manually.

### 4.1. Images Dataset

To generate this dataset, a virtual office environment in ROS/Gazebo (Figure 4) was used. For each emotion, six groups of three members were formed, and the trajectory of the robot where faces were detected at different angles and directions was recorded. Then, the detected faces were automatically stored and tagged for each emotion. Hence, approximately 4000 faces were generated for each emotion, from which 82% were used as training samples and 18% as evaluation samples. In total, this dataset contains 23,222 images of faces that are classified according to six emotions (happiness, sadness, anger, surprise, disgust, and fear). Table 4 shows the number of images for each emotion and the distribution for training and testing.

### 4.2. Videos Dataset

To generate this database, two virtual environments of ROS/Gazebo were used: a museum and a cafeteria (Figure 5 and Figure 6, respectively). In this case, the important issue is to form groups of people who have different emotions. In each virtual environment, 12 groups were formed, and how the robot moves forward and sweeps to capture all the faces in the group were recorded. Each video consists of approximately 60 frames and is approximately 4 s long. Then, there was manual tagging of the emotion for each frame.

For the formation of the groups in the videos, five people were used for each emotion, that is a representation of 30 people in total. These representations of people are different from those used in the creation of the image database. The formation of groups was carried out considering the circular formation (groups of three people or more) and the side-by-side formation (groups formed by two people). In total, there were 24 videos that were used as test the data to validate the emotion of each frame.

## 5. Simulations and Results

To validate and evaluate the performance of this proposal, A set of experimental simulations was performed. In this section, we present such experiments, as well as the results obtained.

### 5.1. Simulation Environment

For the training of the VGGFace neural network, Google Collaboratory Pro was used (https://colab.research.google.com/, accessed on 1 May 2022), with the “NVIDIA Tesla K80” GPU. ROS Kinetic and Gazebo 7 were used to simulate the behaviour of robots in indoor virtual environments; this version of ROS works with Ubuntu 16.04 and Python 2.7. Furthermore, to implement the proposal of this work in ROS, a virtual environment was created in PyCharm, with the Keras 1.2.2 and Tensorflow 0.12.1 packages. All simulations were performed on a desktop PC, with 16 GB RAM memory, an AMD A10 7860k 4-core CPU, and an AMD RX-570—4 GB graphic card (the GPU was only used to simulate the virtual environments in ROS; no additional drive was installed).

### 5.2. Individual Emotion

To validate the results of the detection of individual emotions, the dataset of 23,222 images was used, from which 82% were used as training samples and 18% as evaluation samples. Figure 7 shows the loss of the training and validation datasets during 20 training epochs. It was observed that the loss value of the model in training and in the test continued to decrease across the epochs. The average loss value was 1.3246 in the validation data and 1.4056 in the training data, which represent the error rate in the prediction. Additionally, the average accuracy value during training was 0.9719 (97.19%), and the average accuracy value in the validation data was 0.9948 (99.48%) (Figure 8). Problems, such as overfitting or misfitting of the data, are not observed in Figure 7 and Figure 8; this indicates that the training process was correct.

Figure 9 shows the confusion matrix of the test data. The predicted labels by the model are represented on the *x* axis, and the true labels are on the *y* axis. The values on the diagonal of the confusion matrix are the predictions made correctly, and the labels that obtain the most correct predictions are represented by blue cells. The confusion matrix shown in Figure 9 was designed with 750 validation images for each label, except the disgust label, having 566 images. Finally, the validation process had only nine incorrect images.

### 5.3. Emotion of Videos

To validate the results of emotion detection in videos, the dataset of 24 videos with two groups of people forming two scenes was used, 12 videos recorded in the virtual museum and 12 videos recorded in the virtual cafeteria. Figure 10 and Figure 11 show the emotions in each video. At the left end are the videos (Video 1, Video 2, *…*, Video 12), and at the right end are the detected emotions. The size of the representative points determines which scene the frame belongs to, and the colour of the points determines the emotion of the frame.

#### 5.3.1. Videos Recorded in the Museum

To validate the results of the detection of emotions in the museum, 12 videos were recorded. Figure 10 shows that six videos were made up of two scenes, and the other six contained only one scene. The emotions detected per frame correspond to what the robot was observing. For example, Video 2 is mostly made up of people with a positive emotion, but it is shown that in Scene 2, there are frames with negative and neutral emotions. This limitation is due to the fact that the robot detects only the emotion that it is seeing, and if this detection is prolonged, it can be the emotion of the scene even if this emotion is not the dominant one.

Table 5 shows more details of the results shown in Figure 10. The less accurate results shown in Figure 10 were due to the lighting inside the virtual museum, and in some cases, the detected faces were not looking directly at the robot’s camera. For example in Video 3, the virtual museum was dark, and in the case of Video 4, the detected faces did not look directly at the robot’s camera. The lowest accuracy was found in Video 3; this is because the dark lighting in the environment caused the face detector (i.e., Viola–Jones model) to consider other regions as faces. These are examples of how the robotic perspective and the environment conditions impact the process of emotion recognition.

#### 5.3.2. Videos Recorded in the Cafeteria

To validate the results of emotion recognition in the virtual cafeteria, 12 videos were recorded. Figure 11 shows that there were four videos made up of one scene (Video 2, Video 8, Video 9, Video 11). This is because there was no substantial change in the area of the detected faces due to the height of the people or the approach of the robot. This would be another example where the robocentric perspective affects the outcome.

Table 6 shows the results obtained with the videos recorded in the cafeteria. The lowest accuracy was found in Video 10; this is because not all faces were detected in the video. In this case, the group had a neutral emotion and the undetected faces generated an error in emotion recognition. The same happened in Video 6, but the undetected faces did not affect the result because the group had a negative emotion, as well as undetected faces. These results demonstrate, once again, that environmental conditions (e.g., lighting) and the robot’s perspective can affect the final classification.

Table 5 and Table 6 show the percentage of accuracy in the recognition of individual emotions for each frame of the 24 videos. To determine the percentage of accuracy of the emotions recognised in each frame, how many emotions were correctly detected with respect to the emotions labelled in the database was calculated. From the experiments, we obtained an average accuracy of 89.78% and 90.84% in emotion recognition in each frame of the twelve videos recorded in the museum and in the cafeteria, respectively. Accuracy is only observed for emotion per frame, because the emotion of a scene depends on the β parameter, so it cannot be labelled.

### 5.4. Simulation in ROS/Gazebo

In Section 5.2 and Section 5.3, the results obtained in the detection of individual emotions and by scenes were validated using the image and video databases. The group emotion detection algorithm (presented in Figure 1) was also tested directly in the simulated scenarios in ROS/Gazebo with the robot Pepper (i.e., as the real robot is executing the algorithm).

Communication in ROS is basically through nodes. When a message is sent in ROS, it transports the message using buses called topics. Each topic has a unique name, and any node can access this topic and send or receive data through it. In this case, the topic “/pepper/camera/front/image_raw” was used. Figure 12 shows the robot Pepper in a virtual office and the image obtained by the front camera in the robot (as shown in the lower left part in Figure 12).

Figure 13 shows how the robot detects people’s faces, detects individual emotions (starting with the faces on the left: sad, sad, and happy), and shows to which scene the current frame belongs (Scene 1), the emotion of the frame (sad), and the emotion of the scene so far (sad). This simulation was recorded and is available (https://www.youtube.com/watch?v=i63hOTaeu-Y, accessed on 1 May 2022). These tests demonstrate the feasibility of implementing this algorithm in real robots.

## 6. Discussion

This first version of this proposal, as a proof-of-concept, demonstrates the feasibility and efficiency of a robotic system capable of recognising group emotions from interactions with humans, through a robotic perspective. This experience provided the opportunity to extract some current limitations, as well as lessons learned.

### 6.1. Datasets with Robocentric Perspective and Group Emotion Detection

Most studies developed in the context of emotion recognition for social robots base their proposed approaches on datasets with a third-person perspective to train and validate the machine learning models—i.e., datasets are built with emotions detected through human vision or by fixed cameras in a determined place, using this information as the perspective of the robot. The egocentric perspective of a robot can change depending on several aspects, such as displacement, vibration, external agents, circular and angular movements, as well as space conditions, which are part of the natural process of the robot when it is moving around the environment. All these conditions impact the final classification result, as shown in Section 5.3.1 and Section 5.3.2.

Furthermore, most of the available datasets are limited to the recognition of individual emotions, neglecting repositories suitable for group emotion recognition. The robocentric perspective has become a key factor in recent studies for building datasets in other areas, such as human recognition [64], conversational group detection [68,69,70], objects detection [71], and visual–inertial navigation [72,73,74]; however, as far as we know, there are no available egocentric datasets for group emotion recognition. Therefore, the need for datasets of images and videos of groups of people expressing emotions created from sensors available in the robot (camera, Lidar, IMU, encoder, etc.) and with different camera angles, robot joint positions, etc., which reflect the first-person perspective of the robot, is evident. Thus, it is possible to have better training, testing, and validation of group emotion recognition models in social robotics scenarios.

Although the method proposed to create robocentric perspective datasets was developed in virtual environments, the visual perspective of the robot and some aspects related to robots, such as the movement of the robot’s head and its displacement, were considered. A similar method can be implemented in a real robot with real people, which may improve the results, especially considering the aspects of real-time response and characterisation of other robot’s aspects. For future studies in this area, the creation of new datasets that take into consideration the robocentric perspective and suitability for group emotion recognition is essential. Furthermore, to improve accuracy, it is planned to generate other datasets, considering postures, groups, and other objects besides faces, from a robocentric perspective and create a multimodal system. A multimodal system to recognise emotions is intended to simultaneously consider several modalities (e.g., faces, postures, context, voice), since people express emotion in several ways [53,75]. It is expected that researchers that have these possibilities can share their datasets for the community interested.

### 6.2. Emotion of a Scene

In this work, we proposed the idea of the “emotion of a scene” through the recognition of groups of people and their emotions. This concept can be applied in cases in which the context in which the group is acting is relevant, for example recognising the emotion that an artwork (in a museum), a speaker (in a conference), an animal (in a zoo), or food (in a restaurant) produces in a group of people.

In this research, to identify a scene, we only considered the group of people, based on the size of the faces of the people in the groups. Although the results obtained were satisfactory, the scene detection was still limited. Thus, it is planned to extend this approach by considering the context of the group, such as other people or objects near the group, as well as the conditions of the whole scene. Consequently, with the emotions of various scenes, it is possible to determine the emotion of an environment

### 6.3. Applicability

The main objective of this work was to identify the emotion of a group of people, through the perception of the robot. Then, why would a robot want to detect the emotion of groups of people? An obvious application is to design the behaviour of robots and improve HRI to make them more socially accepted. For example, if the robot identifies a negative emotion of the group, it moves away to avoid conflicts or has a submissive reaction; in contrast, if the emotion detected is positive, the robot can approach and talk to the group. Another example of an application is for the robot to have the task of monitoring and registering the emotions of groups of people participating in a course, conference, musical presentation, museum, etc.; this is beyond recognising the emotion of a group, but the emotion of a scene defined for the group; this information can be used not only to define the behaviour of robots and improve HRI, but for post-analysis; secondly, this information can be analysed, classified, and evaluated for further actions and decisions related to the specific environment. Among these examples, many others can arise in the context of modelling the robots’ behaviour, improving HRI, and other aspects in which social robotics can be immersed.

## 7. Conclusions

In this article, we proposed a system to detect group emotions from a robocentric perspective, which can be applied and extended to identify scene emotions. To do so, the VGGFace model was used for individual emotion detection and an emotion fusion was performed to detect group emotion, scene emotion, and video emotion. Due to the lack of suitable datasets for training and testing group emotion recognition models, we also proposed a methodology to generate datasets of images and videos from the egocentric perspective of the robot, i.e., considering the sensory capacity of the robot, which in turn can be influenced by its movement, position, vision angle, etc., and by the environment conditions (e.g., lighting). The proposed approach was proven in several simulated scenarios with a Pepper robot, obtaining results that demonstrated the efficiency and good performance of the proposed system, as well as the feasibility of being applicable in the robotics domain.

The ongoing work is focused on developing the methodology to build robocentric datasets for group emotion recognition in real robots, improving the identification of a scene and its emotion, and implementing the whole group emotion recognition pipeline in real robots to model their behaviours in HRI or social navigation.

## Figures and Tables

**Figure 1 sensors-22-03749-f001:**
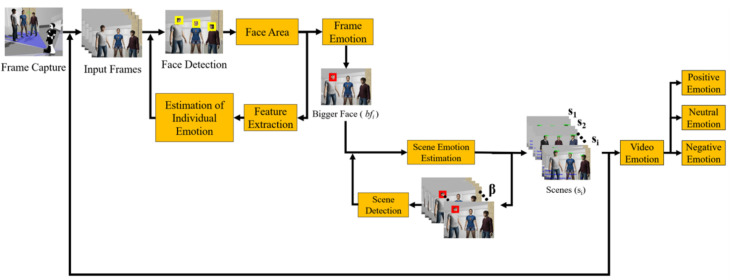
Process for the detection of scenes and the recognition of the emotion of scenes and video.

**Figure 2 sensors-22-03749-f002:**
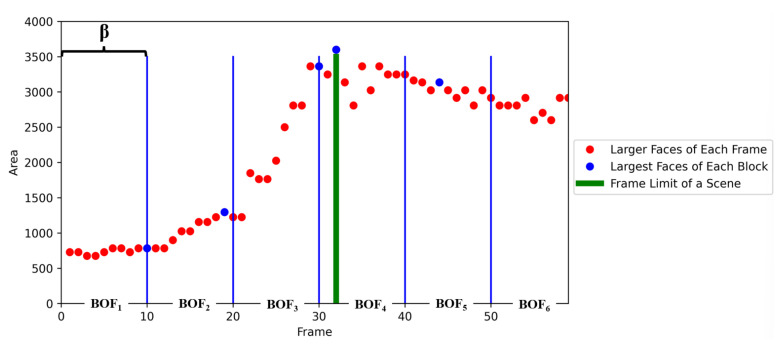
Scene detection with face areas.

**Figure 3 sensors-22-03749-f003:**
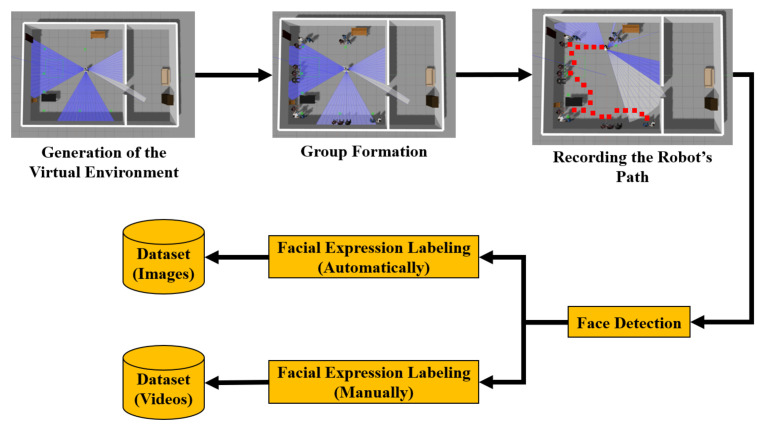
Applied method to create the datasets.

**Figure 4 sensors-22-03749-f004:**
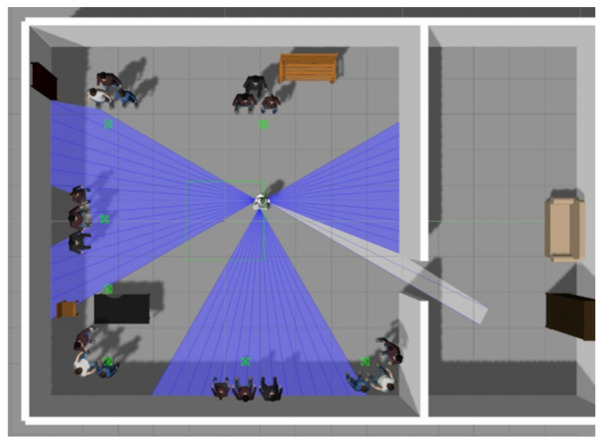
Virtual office used to generate the image dataset.

**Figure 5 sensors-22-03749-f005:**
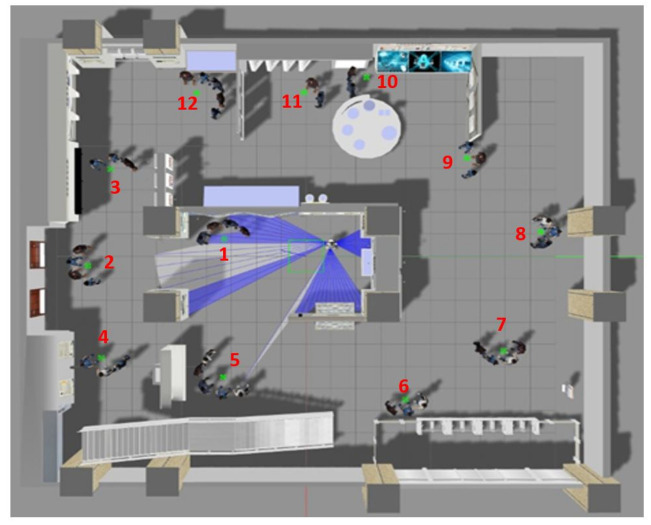
Virtual museum used to generate the 12 videos conforming the videos dataset.

**Figure 6 sensors-22-03749-f006:**
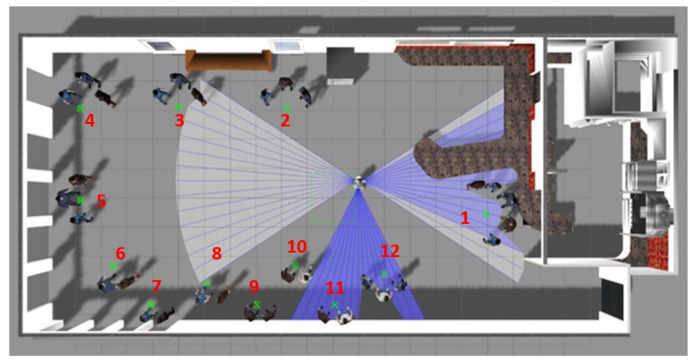
Virtual cafeteria used to generate the 12 videos conforming the videos dataset.

**Figure 7 sensors-22-03749-f007:**
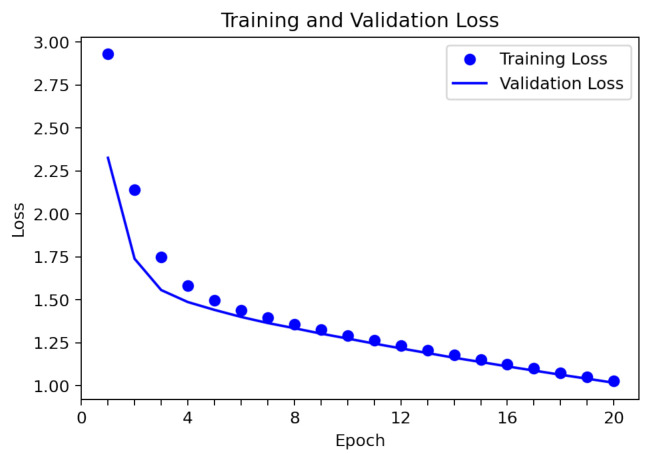
Loss of the modified VGGFace neural network.

**Figure 8 sensors-22-03749-f008:**
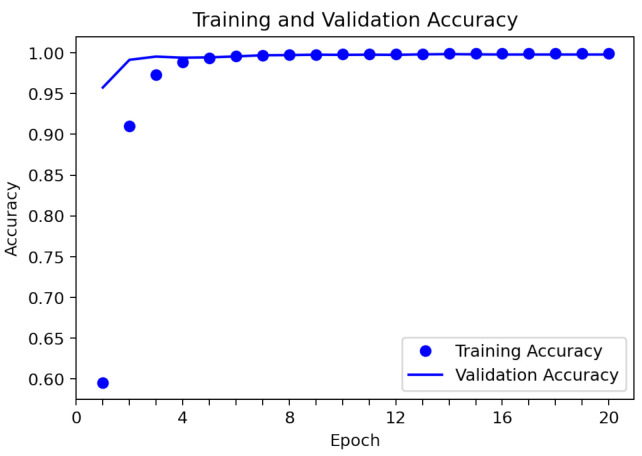
Accuracy of the modified VGGFace neural network.

**Figure 9 sensors-22-03749-f009:**
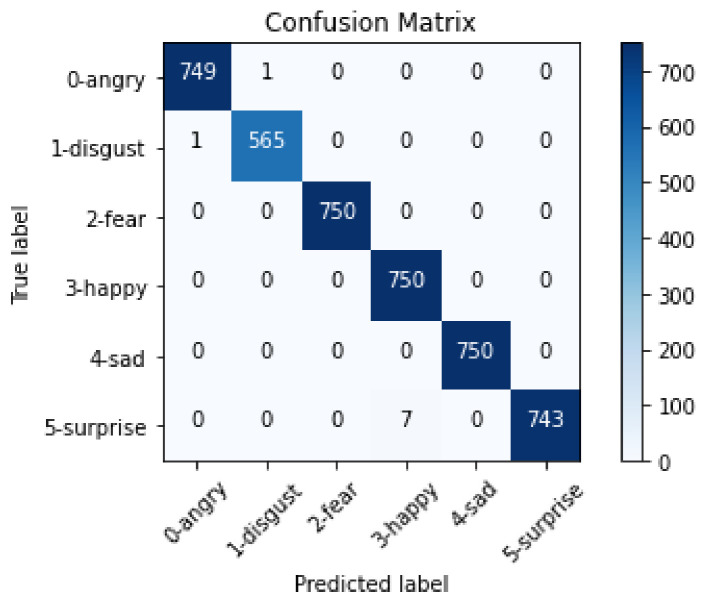
Confusion matrix of the modified VGGFace neural network.

**Figure 10 sensors-22-03749-f010:**
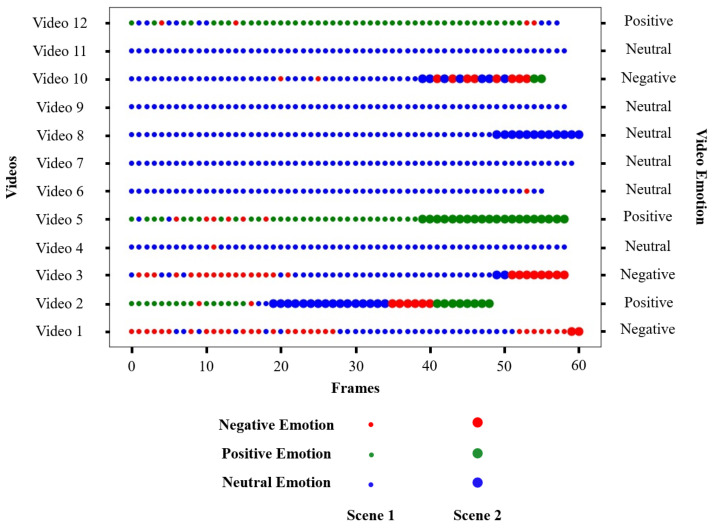
Results obtained for each video recorded in the museum.

**Figure 11 sensors-22-03749-f011:**
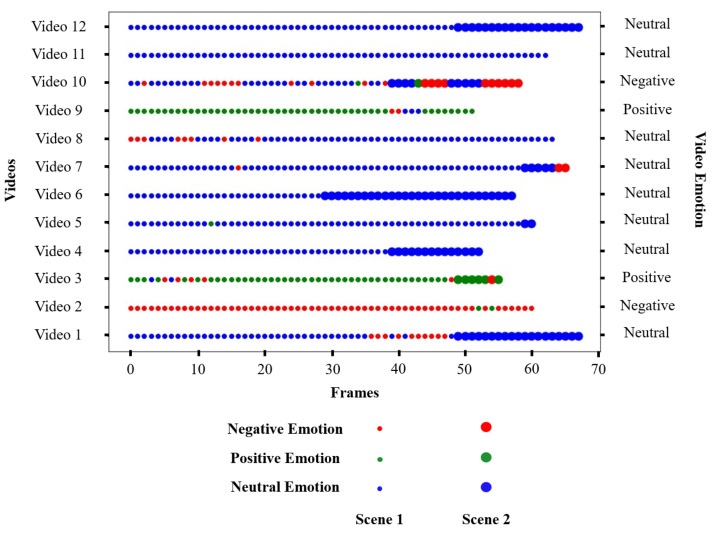
Results obtained for each video recorded in the cafeteria.

**Figure 12 sensors-22-03749-f012:**
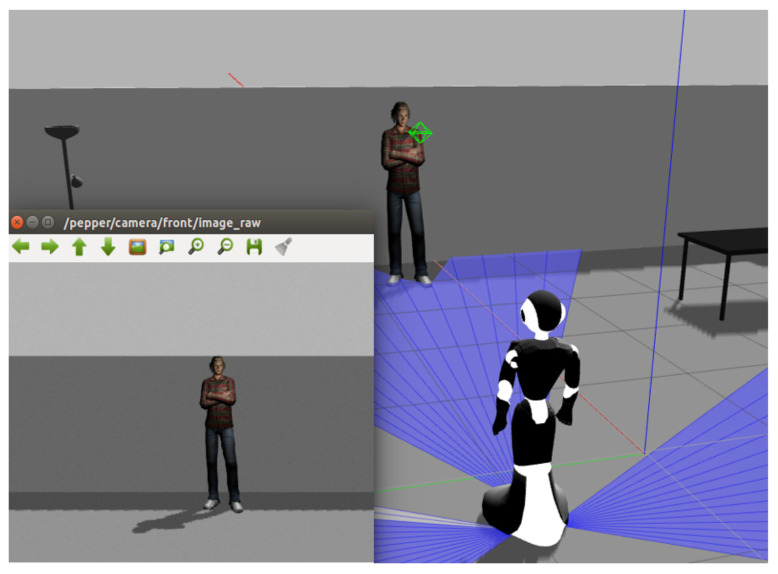
Image obtained by the front camera of the Pepper robot.

**Figure 13 sensors-22-03749-f013:**
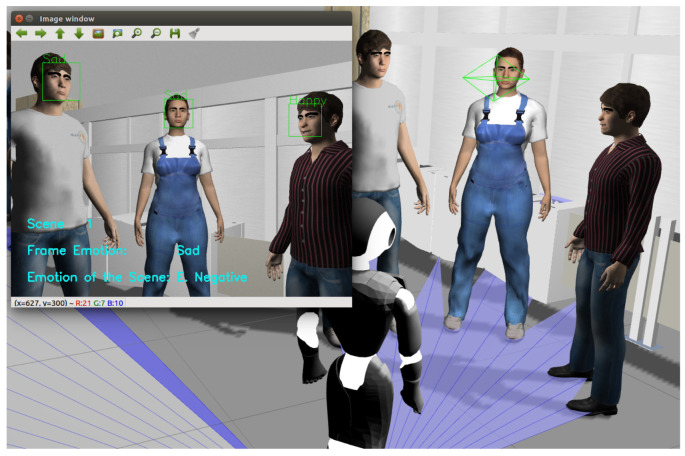
Emotions detected in the first scene.

**Table 1 sensors-22-03749-t001:** Comparison of methods for individual emotion recognition.

Reference	Pre-Processing for Face Detection	Individual Emotion Detection Model	Fusion Method
Sun et al., 2016 [36]	Intraface	AlexNet	LSTM
Tan et al., 2017 [24]	MTCNN	ResNet-64 and ResNet-34	Average
Guo et al., 2017 [17]	Regression Trees and Viola–Jones	VGGFace	Weighted Sum
Wei et al., 2017 [35]	Seetaface	VGGFace with LSTM and DCNN with LSTM	LSTM
Rassadin et al., 2017 [34]	HOG and Viola–Jones	VGGFace	Unmentioned
Abbas and Chalup, 2017 [37]	Mixtures of Trees Method	CNN	Unmentioned
Balaji and Oruganti, 2017 [31]	TinyFace	VGGFace	Unmentioned
Guo et al., 2018 [18]	MTCNN	VGGFace and VGG2-SENet-ft-FACE	Weighted Sum
Wang et al., 2018 [25]	MTCNN	ResNet64, VGGFace, ResNet-34 and SENet154	Cascade Attention Networks
Khan et al., 2018 [26]	MTCNN	ResNet-18 and ResNet-34	Average
Gupta et al., 2018 [27]	MTCNN	Deep Hypersphere Embedding for Face Recognition	Attention Mechanisms
Xuan Dang et al., 2019 [19]	PyramidBox and TinyFace	ResNet50, Inception-ResNet-v2 and DenseNet201	Combination of Feature Vectors
Guo et al., 2019 [32]	S3FD and MTCNN	ResNet18	Cascade Attention Networks
Zhu et al., 2019 [23]	MTCNN	VGGFace	Average
Yu et al., 2019 [40]	Unmentioned	VGG-16, MobileNet-v1	Bi-directional LSTM
Guo et al., 2020 [28]	MTCNN	VGGFace and GNN	Unmentioned
Sun et al., 2020 [30]	RetinaFace	ResNet and BNInception	FAN Model
Tien et al., 2021 [20]	TinyFace	ResNet50	MLP network with D2C block
Khan et al., 2021 [29]	MTCNN	VGGFace and GNN	Context-aware Fusion
Quach et al., 2022 [21]	RetinaFace	EmoNet	NVPF

**Table 2 sensors-22-03749-t002:** Fully connected layer settings.

Layer (Type)	Output Shape
fc8 (Dense)	(None, 512)
d1 (Dropout)	(None, 512)
fc7 (Dense)	(None, 512)
d2 (Dropout)	(None, 512)
fc6 (Dense)	(None, 6)

**Table 3 sensors-22-03749-t003:** Classification of the six basic emotions.

Positive Emotions	Neutral Emotions	Negative Emotions
Happy	Surprise	Sad, Fear, Disgust, and Angry

**Table 4 sensors-22-03749-t004:** Images dataset.

Emotions	Happy	Sad	Angry	Disgust	Surprise	Fear
Training Data	3371	3145	3363	2872	3179	2976
Test Data	750	750	750	566	750	750
Total	4121	3895	4113	3438	3929	3726

**Table 5 sensors-22-03749-t005:** Summary table of the emotions detected in each video and the accuracy of emotion detection in each frame for the videos recorded at the museum.

Video	Emotion of the Scene	Emotion of the Video	Accuracy
Video 1	Neutral Emotion	Negative Emotion	1.0000
	Negative Emotion		
Video 2	Positive Emotion	Positive Emotion	1.0000
	Neutral Emotion		
Video 3	Neutral Emotion	Negative Emotion	0.5094
	Negative Emotion		
Video 4	Neutral Emotion	Neutral Emotion	0.7736
Video 5	Positive Emotion	Positive Emotion	0.8113
	Positive Emotion		
Video 6	Neutral Emotion	Neutral Emotion	0.9811
Video 7	Neutral Emotion	Neutral Emotion	1.0000
Video 8	Neutral Emotion	Neutral Emotion	0.9811
	Neutral Emotion		
Video 9	Neutral Emotion	Neutral Emotion	1.0000
Video 10	Neutral Emotion	Negative Emotion	0.7924
	Negative Emotion		
Video 11	Neutral Emotion	Neutral Emotion	0.9811
Video 12	Positive Emotion	Positive Emotion	0.9434
Average Accuracy	-	-	0.8978

**Table 6 sensors-22-03749-t006:** Summary table of the emotions detected in each video and the accuracy of emotion detection in each frame for the videos recorded in the cafeteria.

Video	Emotion of the Scene	Emotion of the Video	Accuracy
Video 1	Neutral Emotion	Neutral Emotion	0.9559
	Neutral Emotion		
Video 2	Negative Emotion	Negative Emotion	1.0000
Video 3	Positive Emotion	Positive Emotion	0.7544
	Positive Emotion		
Video 4	Neutral Emotion	Neutral Emotion	1.0000
	Neutral Emotion		
Video 5	Neutral Emotion	Neutral Emotion	1.0000
	Neutral Emotion		
Video 6	Neutral Emotion	Neutral Emotion	1.0000
	Neutral Emotion		
Video 7	Neutral Emotion	Neutral Emotion	1.0000
	Neutral Emotion		
Video 8	Neutral Emotion	Neutral Emotion	0.7813
Video 9	Positive Emotion	Neutral Emotion	0.9811
Video 10	Neutral Emotion	Negative Emotion	0.4286
	Negative Emotion		
Video 11	Neutral Emotion	Neutral Emotion	1.0000
Video 12	Neutral Emotion	Neutral Emotion	1.0000
	Neutral Emotion		
Average Accuracy	-	-	0.9084

## Data Availability

Data available in a publicly accessible repository.

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
