# Peer review of "Group Emotion Detection Based on Social Robot Perception"

_sensors, 2022, doi:10.3390/s22103749_

Round 1

Reviewer 1 Report

Our peer review touches some parts of your article, which had to be revised before final publication. We will split our review into two parts for ease of use: issues and suggestions.

Suggestions:

On line 5 of the original article, in part Abstract, we suggest replacing word positive to more neutral synonyms as “desired” or “needed”. In general, if speaking about human-robotic interaction, it’s acceptable, but in context of the article we suggest replacing it with other term.

On line 15 in Abstract, we suggest to rephrase term “each frame of the scene” with more specific term “each frame of the video stream of visual sensor, observing the virtual scene”.

On line 25, we suggest adding more key words like: “emotion recognition” (or “facial expression recognition”, “group behavior recognition”, “human-robot interaction”.

On line 30 we suggest substituting “mimic” with “imitate”.

On line 130 we suggest adding disambiguation of word “visual attentions” in context of following paragraphs or using a substitute word having same meaning, for instance “visual attentions (points of interest of members of the group)” since term “attention” can be mistaken by readers with term from computer science, e.g. “Cascade Attention Network”.

On line 200 we suggest combining the next two sentences in one using separators “:”, “;” etc.

On line 211 we suggest to replace term “adapt” in phrase “adapt behaviors” with other synonymous words “create”, “generate”, “formulate” or others, since you create synthetic behavior of virtual robotic system and define its interactions.

On line 214 we suggest removing term “emotional intelligence” (E.I.). Since E.I. is a complex term, containing such properties as human-to-human interaction ability, ability to feel empathy to another human being and many others, for a state-of-art human-computer interaction systems (or human-robotic ones), which are the case of this study, it can’t be applied.

On line 219 one can rewrite “proposals” with “research papers”, as done in sentence on line 232.

On line 228 we suggest substituting phrase “they can estimate the best routes to follow” with phrase “robots can plan the best routes to follow”, since the phrase has connection to terms “robots” and “people”, so as it can be misunderstood, also “planning” is better suited in context of robotics.

On line 240 we suggest combining the following sentences in one using numbered list, for instance “reports, such as: 1) … states that …; 2) … made this possible by introducing …; 3) … improved the solution from …”. Simple sentences, inheriting same subject from main sentence are easier to read if re-written in such manner.

On line 248 it is better to replace “several” with “multiple”.

On line 441 it is rather better to uses “series” or “a set” instead of “battery” since this term (in context) can be not very well-known for wide auditory of readers of the article.

On line 446 we suggest adding reference to a Google Collab site (hyperlink) and noting, if possible, which Google GPU was used in these experiments.

On line 450 you note, that your test system utilizes TensorFlow 0.12.1, which is, obviously, mistaken with library for Radeon GPUs (you use Radeon RX 570). Please specify which one (PlaidML, RocM).

On line 460, we strongly suggest, like on line 512, write down the average accuracy instead of maximum achievable. It gives you a row for an error in experiments (even if there is not), since such big accuracy in general purpose machine learning and using real-life data can’t be easily achieved using such data.

On line 487 and 489 we suggest using “dark”, “obscured”, “not fully set” instead of “very bad” since it’s a properties/conditions of the scene (lighting) and not a drawback of your experiment.

Issues:

On line 12, in Abstract single sentence has multiple issues.

  • We suggest (just for your readers) add disambiguation for word “model” and specify what type of model it is – e.g. imitational, physical, mathematical, computer, animation etc.
  • Also in the same sentence on the line 12, we suggest replace phrase “to recognize the scenes” with phrase, more specific in context of your article “to visualize and recognize emotions in typical robot-human interactions”, since it can more easily explain task of your research and your scientific goals. The same can be said about the beginning of the sentence on line 13.
  • Also, the same sentence contain term “emotion emotion in the scene”, which, in context of your article has to be replaced with more specific term “global (prevalent) emotion in the scene”. The same can be said about the sentence on line 16.

On line 21, in Abstract, the beginning of the sentence contains a typo.

On line 40, in Abstract word “specif” contains a typo.

Sentences on line 68-87 in chapter introduction are very alike to some sentences in abstract; for instance, sentence on line 68 follows the meaning sentence on line 12, sentence on line 71 follows the meaning sentence on line 13. Beyond the repetition of the meaning of sentences from abstract in introductory chapter, part of them (sentences on line 83-88) can’t be presented in introduction, only in “results”, since it can’t contain the results of the following article, with exception of previous results, but placed in the paragraph/part dedicated to background/previous works in area of research.

Line 163 possibly contains a typo in word “A-Ssoftmax”.

On line 240 the name of computation device Kinect has to be written from the capital letter.

On line 425 text “Table ??” should be rewritten with “Table 4”.

On line 612 one should rewrite the names of authors with full last (family) names.

According to our observations, your experimental and theoretical (in particular algorithm specification and denotion of terms) part is very strong; same can be said about literature survey. Although, in order to be published, we strongly suggest you to solve all issues and take in account our suggestions about your article, also the abstract and introductory chapter (first 100 lines of article) need serious rework.

Reviewer 2 Report

The article “Group Emotion Detection Based on Social Robot Perception” presents a system to recognize emotion in a group of people in a video, detecting faces in the frames, recognizing individual emotions and averaging them to obtain the emotion of the group. System considers images get by the robot camera and uses the growing or decreasing of face biggest area to determine when a scene is changing. Results presented are based in images obtained by a robot Pepper in a simulated environment where people with fixed emotions are inserted forming groups of two aligned or three in circular position. Authors present their dataset composed of pictures taken with the robot simulated in an office simulated environment and videos obtained in a simulated environment of a museum and a cafeteria as a contribution of this work.

Besides the difficult of constructing simulated environments and the quality of scenarios presented, it is totally different from realizing real experiments. Images obtained from a robot camera when it is moving and the effects of the movements in the images cannot be represented. Considering faces with fixed emotions along the time is out of reality, since real faces change a lot in the time when people talk or even if they are seeing a move or a presentation.

The article is well written and easy to read, but there are some points to correct to improve it that are listed in the following.

Considering the email [email protected] it is possible to infer a typing error in the author’s name Raquel Patiño.

At line 21 in the abstract “Test are performed in two” it should be “Tests are performed in two”.

It is missing a reference or (even a footnote with a site) for VGGFace neural networks at line 74. What does VGG mean?

At lines 84 and 85 there is a confusion about the environment simulated: “several experiments are carried out with two simulated scenarios: an office and a museum”. In the article are presented results from two scenarios: a museum and a cafeteria.

Presentation of results in the introduction at lines from 85 to 90 are not correct. They should be only at results section.

The format of quotation marks at lines 130, 525 and 569 are different from each other.

The format of regions of images at lines 154 (48x48) and 159 (40*40) are different.

What is the correct form to write: “pre-trained” or “pretrained”? Text has both forms.

Typing mistake at line 163: “(A-Ssoftmax)” should be “(A-Softmax)”.

Typing mistake at line 166: “covolutional” should be “convolutional”.

ResNet is presented as abbreviature of Residual Network at line 180, but it is used before.

At Table 1 there are acronyms not defined before.

At line 244, what does NLP mean?

At lines 268, 269 and 270, “The RICA database [64], a database generated from a robocentric view, was found, but it was focused on group iterations”. Does the sentence refer to group iterations or interactions?

At line 274 “Viola Jones algorithm” should be changed to “Viola-Jones algorithm”.

At lines 302 and 303, the sentence “A BOF can contain two scenes at maximum or several BOF might belong to the same scene” sounds incorrect considering that size o BOF is defined by the user through parameter beta.

At Figure 1 it should be interesting show where flow starts in the diagram, because flow is commonly represented from left to right and the emotion output is on the left end.

At lines 307, 308 and 309, “On each frame with people, the face detection algorithm is carried out, the area of each face is calculated, the extraction of characteristics is performed, and the individual emotion of each face in the frame is estimated.” There is a loop in the diagram on Figure 1 involving this part, where faces are detected, characteristics are extracted and emotion is recognized. However, it is not clear if the algorithm detects all faces in the image at the same time, when it decides the number of faces and how face areas are separated to give information for emotion detection.

At line 331 Section 3.1. Face Detection: “The objective of this stage is to obtain the area of each face, whose information is used for the scene detection. To do so, we use the Viola-Jones classifier as a face detector.” Once again, it is not clear how Viola-Jones brings information about multiple faces.

At line 345 Section 3.3. Estimation of Individual Emotion: “VGGFace was trained to recognize 2,622 classes. However, in this case we do not have 2,622 emotions to classify; therefore, we modify the fully connected layers of the VGGFace model, shown in Table 2. Layers fc6 and fc7 have 512 nodes and layer fc8 has 6 nodes, which represent the emotions to be classified (happy, sad, angry, fear, disgust, and surprise).” When VGGFace architecture is changed it has to be trained again. It should be informed here. Using artificial images generated in a simulated environment in a neural network trained with images from real scenarios may be a problem too. It should be commented in the text.  

The variable l is defined at lines 302, 303 e 304 in a range from 1 to 2k. In the Algorithm 1 l is used to iterate between 1 to beta (line 4). It should be interesting using another variable considering that they are not the same.

At Algorithm 1, lines 1, 11, 12 and 13, f1 and f2 are used to designate first frame and last frame respectively in a sequence that corresponds to a scene. It should be more interesting using another notation (for example, finit and fend) since frames are numbered for other reasons.

It is missing a reference at line 394 to EmotiW dataset.

At line 425 it is missing a table number: ”Table ?? shows the number of images for each emotion and the distribution for training and test.”

Table 4 presents typing mistakes in the title (“Datset” for “Dataset”) and first column (“trainig” for “training”).

Vertical axis at Figure 8 presents typing problems: “Acuracy” for “Accuracy”.

At lines 467, 468 and 469: “The confusion matrix shown in Figure 9 was designed with 750 validation images for each label except the disgust label with having 566 images.” It should be changed for “The confusion matrix shown in Figure 9 was designed with 750 validation images for each label except the disgust label with 566 images.” or “The confusion matrix shown in Figure 9 was designed with 750 validation images for each label except the disgust label having 566 images.”

Are the results at Figure 10 and Figure 11 considering that the robot faced only two scenes? It is not clear in the text.”

At lines 522 and 523: “When a message is sent in ROS, it transports it using buses called topics.” It should be changed to ““When a message is sent in ROS, it transports the message using buses called topics.”

At lines 564, 565 and 566: “For future studies on this area, it is a must the creation of new datasets which takes into consideration the robocentric perspective and suitable for group emotion recognition.” It should be changer to: “For future studies on this area, it is essential the creation of new datasets which takes into consideration the robocentric perspective and suitable for group emotion recognition.”

At lines 583 and 584: “An obvious applicability is to design the behaviour of robots and better HRI to make them more socially accepted.” It should be changed to: “An obvious applicability is to design the behaviour of robots and improve HRI to make them more socially accepted.”

At line 585 “it gets away to avoid conflicts or have a submissive reaction” should be changed to “it gets away to avoid conflicts or presents a submissive reaction” or “it gets away to avoid conflicts or has a submissive reaction”.

Reviewer 3 Report

The paper "Group Emotion Detection Based on Social Robot Perception" presents a new method to detect scenes according to face recognition. Emotions are detected using neural networks. The tests were performed in a museum and in a cafeteria.

- Originality/Novelty: The authors provide a novel method for a robot to perceive the environment based on the emotions of the people who are around it.

- Significance: The results of the research are interpreted properly, along with tables and figures. 

- Quality of Presentation: 

The article is written appropriately, respecting the logical succession of sections. Data and analyses are presented graphically and inside tables. 

Figures 1, 3 should be enlarged, such that the readers can visualise them better.

Based enrich table 3 regarding basic emotions. Please study Plutchik's wheel of emotions and refer to it.

At line 425 there is no table number. You should mention that it's in fact table 4.

Which program was used to create the 3D space of Figure 5 and 6? It should be mentioned in the paper, in the section where they appear. Afterwards it is discovered that the ROS / Gazebo tool was used for them.

How do you detect if happiness is mimicked? A person can smile for a second, just to be kind and then they change their face appearance.

Please avoid using the pronoun we excessively - "In this article we propose a .... To do so, we use... We proved our approach...". Please rewritten such phrases.

As future work, how do you think that a better accuracy can be reached?

- Scientific Soundness: 
The findings and their implications should be discussed in the broadest context possible and limitations of the work should be also further highlighted.

The conclusions and discussions section should be enriched to provide more details about the contribution of their study to existing literature.

- Interest to the Readers: The conclusions would surely interest the readers of the Sensors journal, and not only them, as emotion detection is in trends. The paper is interesting and will attract many researchers.  

- Overall Merit: The described solution can be implemented by other researchers and can be involved in other experiments. 

- English Level: The level of English language is advanced. Through the entire paper, the language was appropriate and understandable, being easy to follow the flow since the beginning.

Round 2

Reviewer 1 Report

In general authors in very correct form given answers on the all my remarks. I think that article look more better and I recommend to accept ones in this view.